

# Optimization of the WRFV3.7 adjoint model

Qiang Cheng[1], Juanjuan Liu[2,4], Bin Wang[2,3,4]

[1]School of Computer & Information Sciences, Southwest University, Chongqing, 400715

[2]LASG, Institute of Atmospheric Physics, Beijing 100029, China

[3] Ministry of Education Key Laboratory for Earth System Modeling, Department of Earth System Science, Tsinghua University, Beijing 100084, China

[4]University of Chinese Academy of Sciences, Beijing 100049, China

*Correspondence to:* Juanjuan Liu (ljjxgg@mail.iap.ac.cn)

**Abstract.** This work focused on a new strategy for productively improving the performance of adjoint models. By using
several techniques including the push/pop-free method, careful Input/Output (IO) analysis and the use of the conception of adjoint locality, we reduced the adjoint cost of the Weather Research and Forecasting plus (WRFPLUS) by almost half on different numbers of processors especially with a slight decrease in total memory. Several experiments are conducted using the four-dimensional variational data assimilation (4DVar) method. The results show that the total time cost of running a 4DVar application is decreased by approximately 1/3.

## 1 Introduction

Currently, adjoint models required in many large-scaled nonlinear optimization applications (Courtier and Talagrand 1987, Charpentier and Ghemires. 2000, Vidard et al. 2015, An et al. 2016) have been developed using scores of automatic differentiation (AD) tools in an automatic way (Giering 1997, Giering et al., 1998). However, such models are not always productive for a specific large application, either in computational cost or in correctness. Although the true computational cost

of the adjoint model has been said to be no more than five times of that of the underlying functions of two decades ago (Griewank 1989), this ratio related to the adjoint model generated with AD tools could range from several to scores in practical applications (Giering et al. 2005, Heimbach et al. 2005, Cheng et al. 2009b, Hascoet and Pascual 2013, Guerrette and Henze 2015). Generally, several factors result in such huge computational cost, such as the lack of the knowledge of required variables, inflexible adjoining strategies, inefficient push/pop implementations, poor locality of data references, and large number of

redundant recalculations.

Regarding the raw codes generated with AD tools only, much work remains after checking their correctness. First, there are many unnecessary push/pop operations and unnecessary calculations inside them stemming from the lack of necessary IO knowledge for AD tools. Second, it is not wise for some adjoint codes such as generated using TAPENADE (Hascoet and Pascual 2013) to first calculate each partial derivative item and then accumulate them by multiplying their respective



perturbations during local adjoint accumulations. Apparently, this could require more unnecessary recalculations for the same intermediate results. Next, we have found that some extra costs of most adjoint implementations are due to the large number of push/pop operations. In fact, most of them can be removed or replaced in more productive ways closely related to the recognition of the "locality" characteristics of adjoint/reverse accumulations. In addition, each successful adjoint implementation of a specific application must have both an acceptable computational cost and acceptable memory consumption to the users. To some AD applications such as 4DVAR, much more memory consumption must result in poor scalability both in computing grids and in different computer environments.

An adjoint calculation at a current position is in some way independent of the calculations in other places. This is called "adjoint locality" in this article. In fact, we can obtain the required data for the current adjoint accumulations in any way (Griewank 2000, Giering and Kaminski 2002, Hascoet et al. 2005). For example, if the value of a variable within a subroutine is required for calculating a partial derivative at current position, we can obtain this value, which is called the basic state value in some documents, in different ways such as directly recalculating it from the start of the underlying functions, calculating it from intermediate results, or directly restoring it from the stack in which it has been kept in the last run of this subroutine where there is often far more than one place at which the stack operations can be inserted. However, all of these operations will never be a disadvantage to the proper results of the adjoint model because of such independency between these adjoint calculations. Apparently, different ways have different costs in run time or memory. Moreover, this conception is necessary not only in the calculation of local partial derivatives but also in adjoint calculations within other program objects such as a segment of program lines, a loop or switch structure, a subroutine or function, or a module.

The Weather Research and Forecasting (WRF) model is a numerical weather prediction and atmospheric simulation system designed for both research and operational applications (Skamarock et al. 2008) that has thousands of users worldwide. The adjoint model and the tangent linear model within WRFPLUS are used to productively calculate gradients of a cost function in its 4DVar system (WRFDA) and an adjoint integration is required once per 4DVar minimization iteration step (Huang et al. 2009). For a more realistic configuration, the minimization stops after about 30 to 50 iterations (Vidard et al. 2015). Thus, even for a middle-scaled application, the time cost of the adjoint model could be more than half of the time required to run the total 4DVar system. The core subroutine of the adjoint model, which is executed scores of times in each Newtonian iteration step, is simply implemented in a PUSH/POP strategy that results in huge memory requirement in large-scaled applications. Moreover, such a strategy is inefficiently used in adjoint implementations. Therefore, it is necessary to improve the adjoint model with respect to both computational time and memory.

The tangent linear model and the adjoint model within WRFPLUS and WRF data assimilation system (WRFDA; Barker et al. 2004, 2012) have been coupled to execute the 4DVar system. The updated tangent linear model and the adjoint model of WRF (Zhang et al. 2013) were mainly established with DFT/ADG (Cheng et al. 2009a, b) and TAPENADE by a slight of hand several years ago. Both tools can be used to generate adjoint codes with different adjoint strategies. In this application, each adjoint subroutine/function produced with TAPENADE is made using a simple push/pop strategy, which requires less computational cost but much more memory consumption. Actually, the computational cost for these push/pop operations will

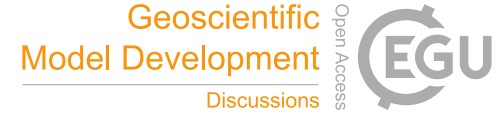



turn out to be comparatively expensive when the local adjoint cost is reduced at a lower degree. However, the adjoint codes generated with DFT/ADG are implemented in a least program behavior decomposition method (Cheng et al. 2004) in which a subroutine/ function is first split into several small partitions according to their specific program structure. The advantage of this implementation is that extra memory space can be shared across these partitions, and the number of dimensions of each

saving variable is often reduced by one compared with its corresponding required variable. This implementation requires comparatively less memory consumption but much more computational cost, although there are always productive ways available for reducing the latter in practice.

However, the most difficult optimization may be associated with huge memory consumption stemming from the fact that the computational cost of the core adjoint procedure is almost evenly distributed across its calling subroutines, and the

computational cost of each called subroutine is no more than 10% of the total. This difficulty could lead to a huge memory requirement if the least adjoint running cost is definitely pursued. Additionally, scores of iterations of the core adjoint procedure are required in each calculation of the gradient of the cost function. Assuming that the value of a variable is required during the calculation of the core adjoint procedure, the memory required to keep these values will be scores of times that of the memory of the variable itself if all its values are kept in advance. Therefore, the greatest challenge of this work is

determining how to significantly reduce the running cost without requiring extra memory consumption.

Based on the above discussion, we could first modify adjoint codes by carefully analyzing the global/local IO knowledge for specific variables and removing unnecessary PUSH/POP operations within WRF version 3.7 as well as unnecessary local calculations. Next, we could rearrange some codes by improving their data locality. The conception of adjoint locality and the push/pop free method are proposed for better adjoint performance in Sections 2 and 3, respectively. Some test results are

presented in Section 4, with an emphasis on performance comparison between the underlying and the optimized versions. Conclusions are given in Section 5.

## 2 Adjoint Locality

A program object can be a statement, a segment of statements, a loop or switch structure, a subroutine or a function defined by its inputs/outputs and its calculation process. Mathematically, a program object can be taken as a vector-valued function $F$,

which is a mapping from its inputs to its outputs. For simplicity, we define $\nabla F$ as its Jacobian matrix; therefore, the product $\nabla F \cdot \delta X$ is just the output of the tangent linear model of this program object, and the product $\nabla^{\mathrm{T}} F \cdot \delta Y^*$ is just the output of the adjoint model.

A program object $F$ can be split into a series of much smaller program objects $F_1, F_2, \cdots, F_s$ in some way, assuming that $\nabla^{\mathrm{T}} F_1, \nabla^{\mathrm{T}} F_2, \cdots, \nabla^{\mathrm{T}} F_s$ are their corresponding adjoint implementations. Since the adjoint accumulation is always

performed in a reverse way, we must face a new problem: how can the basic state value at the current position be obtained if this value was never calculated or was ruined in last run of the underlying functions? As a typical strategy of the reverse accumulations in most adjoint implementations, one can successively push the required data $T_1, T_2, \cdots, T_s$ into a huge stack



L in the last run of the underlying functions and naturally pop them out in the reverse adjoint accumulations one by one. Within each adjoint implementation $\nabla^T F_k$, all local adjoint accumulations are executed after a full sweep of the underlying function $F_k$, in which the push/pop operations are made in the same way as mentioned above. This adjoint implementation of the program object is shown in Figure 1.

5    We pick up any two local adjoint implementations $\nabla^T F_j$ and $\nabla^T F_k$, assuming that j<k; in other words, $\nabla^T F_j$ is executed after $\nabla^T F_k$. As mentioned above, the basic states $T_j$ and $T_k$ are required for performing the adjoint implementations $\nabla^T F_j$ and $\nabla^T F_k$, respectively. Note that the implementation of $\nabla^T F_j$ is completely independent from that of $\nabla^T F_k$ in the following aspects:

   a)  The calculation of the basic states within the sweep of $F_j$ is independent from that within the sweep of $F_k$, and vice versa;

10    b)  The calculation of local partial derivatives in the implementation of $\nabla^T F_j$ is independent from that in the implementation of $\nabla^T F_k$, and vice versa;

   c)  The strategies of adjoint implementation for $\nabla^T F_j$ and $\nabla^T F_k$ can be completely different from each other.

All of these aspects of independent implementation make up the conception "adjoint locality." In short, "adjoint locality" means that the adjoint implementation at the current position is completely independent from those at other places, either in 15 the adjoint strategy or in the calculation of basic states. In fact, nothing but the proper calculation of adjoint perturbations is absolutely indispensable at the current position. Therefore, we can obtain the required data for current adjoint accumulations in any possible way, which gives us more chances to find a better adjoint implementation with an acceptable running cost.

Revisiting Figure 1, each partition $F_j$ should be completely executed within the last sweep of the program object $F$ because each output of $F_j$ could be the input of following partitions $F_{j+1}$, $F_{j+2}$, ···, $F_s$, even including the successors of $F$. 20 However, the implementations of $\nabla^T F_j$ ($1 \leq j \leq s$) are independent of each other during the adjoint accumulations of $F$ so that the individual sweep of $F_j$ ($1 \leq j \leq s$) is also independent from each other. Therefore, the full sweep of the partition $F_j$ closely ahead of the adjoint accumulations is not always indispensable since such a sweep is only required for calculating $\nabla^T F_j$ in the proper way. First, only a small essential part of this sweep could be absolutely required for the calculation of $\nabla^T F_j$. Second, the calculating sequence of this sweep can be completely/partly broken down by obtaining values of the required variables in 25 different ways. As a result, this sweep of $F_j$ could be completely/partly removed as well as those inserted push/pop operations. However, the sweeps of such $F_j$ cannot be saved anyway in those checkpointing algorithms (Griewank and Walther 2000, Stumm et al. 2009, Wang et al. 2009) or else the maximum number of checkpoints could not be reached in this way when the available memory is limited. Under the restriction of a given memory consumption, in another words, no one can make sure which adjoint implementation is of lower computational cost either the maximum number of checkpoints or less in combination 30 of this way. Specifically, the adjoint performance can be significantly improved if there is an expensive subroutine that is called within any partition.





To show such advantages, here is an example selected from the variational data assimilation for GPS/MET Rayshooting Model (Zou et al. 2000). In this application the considered underlying subroutine RefGr is called for many times within each sweep of four subroutines from RP to the considered root subroutine RayFind that is expressed as RefGr⊂RP⊂RKS⊂RayEnd⊂RayFind, where "⊂" means that the former subroutine/function is called within a sweep of the latter. Within a sweep of a given subroutine/function, the number of times a program object is run is called the number of "running cycles ($N_{rc}$)." A program object may have different $N_{rc}$ within different subroutines/functions. The $N_{rc}$ of RefGr could be from several to thousands within each sweep of these subroutines. Designed with a 2-level checkpointing structure, the raw adjoint codes were generated via DFT/ADG as shown in Figure 2.

With the similar implementation as shown in Figure 1, the adjoint subroutine Adj_RefGr first contains a full sweep of the underlying subroutine RefGr followed by its adjoint accumulations. Note that there is only one running cycle of RefGr within a sweep of RP such that we can even obtain all required data by simply pushing them into stacks during each sweep of RP when running its adjoint subroutine Adj_RP and popping them out during the adjoint calculations within Adj_RefGr, which really has a very small cost in memory instead of a full sweep of RefGr as shown in Figure 2. However, the sweep of RP within Adj_RP will be completely removed for better adjoint performance including those expensive calling subroutines such as xyz2g, RefCIRA and Int3SL within Adj_RefGr. In Figure 3, we therefore only keep a very small part of the required data directly from the first run of the root subroutine RayFind with the extra memory of $O(N_{rc}·NV)$, instead of recalculating these expensive subroutines while running Adj_RefGr.

We simply keep a part of the output data of each calling subroutine within RefGr so as to remove these expensive subroutine callings as well as remove some expensive intrinsic function calculations such as exp and sin/cos through a slight of modification of the position where the value of the output variable is kept. In Figure 3, detailed implementation of adjoint optimization of this example is presented.

In most adjoint implementations, the checkpointing method is naturally used across subroutines/functions such that RefGr in this example is executed for many times within each sweep of RP, RKS, RayEnd or RayFind which implies the computational cost of this adjoints is of calling structure dependency (Cheng et al. 2004). In this way we have removed such dependency and finally obtained the root adjoint subroutine Adj_RayFind with a lower computational cost than its corresponding tangent linear subroutine Diff_RayFind. Such a technique has also been successfully used in the adjoint optimization of WRF 3.7, even in some implementations of tangent linear codes.

Another beneficial aspect of adjoint locality is that the memory space of either local or intermediate variables, not including the adjoint variables therein, can be shared across each partition since their local values will never be used in the adjoint calculations that follow. In some cases, this sharing can not only save memory but also improve data locality if both the calculations of local required variables and the adjoint accumulations are located inside the same innermost loop.

**3 Push/Pop-Free Method**



Most adjoint models are implemented using the PUSH/POP strategy as shown in Figure 1. Such a simple strategy is also used in most AD tools such as TAPENADE and OpenAD (http://mercurial.mcs.anl.gov/ad/OpenAD). However, the push/pop operations are expensive in some cases such as when the push operations and the corresponding pop operations are located in two different loops. In addition, not all push/pop operations are necessary if the sequence of the underlying calculations can
be completely/partly broken down. Here we improve them in another way.

Let us start with an example where x is a required variable of length N for the following adjoint calculations at the current position. As shown in Figure 1, we can push its value into a huge stack L by inserting an evaluation statement such as "L(pointer:pointer+N-1) =x(1:N)" into a sweep of the underlying functions. At the same time, we can pop its value from the stack by inserting another evaluation statement such as "x(1:N) =L(pointer:pointer+N-1)" into the corresponding adjoint
accumulations. A practical program implementation of this example is presented in Figure 4 in which both the push and pop operations are located in the same subroutine as shown on the left and in two different subroutines as shown on the right. In both cases there are four interested calling subroutines used in this example, where CALCULATE_DATA is used for calculating the current required values of x only, DATA_USE for performing proper adjoint accumulations with these required values, and PUSHREAL8ARRAY for pushing data into a stack and POPREAL8ARRAY for popping data from the stack as
most TAPENADE codes do. Note that x can be a scalar or multi-dimensional array. What is only required in the current adjoint accumulations is the value of x that can be obtained from the stack L. The stack L must be a long one-dimensional (1D) data structure because of different numbers of dimensions of the pushed variables. Apparently, there is a redundant computational cost caused by these factors.

However, such redundancy can be removed by a modified push/pop strategy. First, we use several or more stacks instead
of only one, each of which is still a 1D data structure. It has been shown that this type of data structure has the advantages of smaller access cost and flexible expressions, either in the communications between procedures or in local program calculations. Many stacks have many flexible and independent push/pop implementations that help the communication of required data be able to perform across different subroutines/functions in proper way. Although the cost of allocating/deallocating many dynamical stacks cannot be neglected compared with the costs of these adjoint procedures, we can significantly reduce it by
allocating/ deallocating them outside of the running cycles of the procedures.

Second, consider the case that both the underlying calculations at the current position and its corresponding adjoints are located within the same subroutine or function, in other words, both the push operation and the corresponding pop operation are also located in the same procedure. For example, as shown in Figure 1, local push operations are inserted in a sweep of the underlying function $F_k$ within each adjoint implementation of $\nabla^T F_k$, and the corresponding pop operations are then made in
the same adjoint procedure marked by $< F_k, \nabla^T F_k >$. In such a case, the value of the required variable x is only used locally for calculating other basic states and adjoint accumulations instead of any other calculation outside of this procedure. Therefore, either push operation in the underlying calculations or pop operation in adjoint accumulations can be removed instead of directly accessing the required value into/from the corresponding stack as shown on the left in Figure 5. Within the adjoint subroutine Adj_A, the required variable x is replaced by a segment of stack keepx(pointer:pointer+N-1) both in



CALCULATE_ DATA and DATA_USE. As a result, all push/pop operations can be removed except for some additional calculations of stack pointers. As noted at the outset, such push/pop removing can significantly reduce the running cost of the adjoint procedure that is already lower enough.

However, the above modification will meet with difficulties if x is both an input and output parameter within the calling
subroutine CALCULATE_DATA, which is located in a loop. In this case, one can added the last segment of stack keepx(pointer-N:pointer-1) that contains the current value of x into its parameter list as input data as well as a few slight of changes to this subroutine as required, without any extra cost both in run time and in memory. Specifically, one cannot remove any push operation if the IO status of the required variable x is not clear but we can tackle it similarly as the way shown on the right by passing the values of x into the segment of stack keepx(pointer: pointer+N-1) after the calling of
CALCULATE_DATA.

Another case is that the push operation and the corresponding pop operation are located in two different subroutines or functions in which the push operation cannot be removed since each of the outputs of current underlying calculations may be required in the following calculations. However, we can still remove the pop operation in the same way as shown on the right in Figure 5 and therefore reduce the cost of the original push/pop implementations by roughly half. Note that the pushing
operation PUSHREAL8ARRAY is replaced by a direct evaluation statement from x to keepx with approximately the same cost both in run time and in memory.

The above discussions tell us that the push/pop operations are always redundant, whereas a completely push/pop-free implementation is not easy in most cases. Without doubt, the development of AD software must be more expensive for this method. Aside from removing some push/pop operations whose costs are relatively expensive in some cases, flexible
programming could be necessary to improve the adjoint performance and code readability.

## 4 Test Results

All modification to the code is limited to the core adjoint subroutine solve_em_ad located within WRFPLUS and its calling subroutines including a sweep of the underlying subroutines and the corresponding adjoint subroutines. Through careful IO analysis, we first removed about 1/5 of the unnecessary push/pop operations and changed some of their places within
solve_em_ad, and then modified most of the rest using the push/pop-free method. This method is also extensively used in these calling adjoint subroutines, together with several other techniques based on the adjoint locality such as partly/completely changing the computational sequence/position of the underling sweeps, code refinement of innermost loops and removing sulfurous calculations. In addition, a few pairs of push/pop operations are added, each of which are inserted across two different subroutines.

To verify the improved performance of this work, two groups of experiments are used to compare the improvement version and underlying version. We use a tutorial example of practical WRF 4DVar V3.7 application provided by the National Center for Atmospheric Research (NCAR). Without any physics processes, there is only adiabatic WRF dynamics within the adjoint



model along with simplified surface friction. The WRF 4DVar system is used to calculate the analysis field by assimilating rawinsonde(SOUND), PILOT, PROFILER, surface data from SYNOP, METAR, SHIP, BUOY, aircraft data from AIREP, satellite retrieved wind (GeoAMV) and GPS precipitable water (GPSPW), and GPS refractivity (GPSREF). First, the WRFDA is run on the domain of 96×60 grids with 60−km horizontal resolution and 41 levels. The analysis date is 12 UTC 5 Feb. 2008,

with an assimilation time window of 6 hours and an integration time step of 360 seconds. Detailed configurations of the test can be found at http://www2.mmm.ucar.edu/wrf/users/wrfda/Docs/user_guide_ V3.7/ users_guide_chap6.htm. All experiments were conducted on a cluster system with 250 nodes, each of which has 20 processors.

Table 1 shows the wall-clock time for one-time-step integration with WRFDA V3.7. Compared with its underlying version, the cost of the core adjoint procedure is reduced by approximately 45-50% on different numbers of processors. As we know,

the cost for running the adjoint model takes more than half of the time needed to run the total 4DVAR system, which leads to a significant (approximately 30-35%) reduction in the running of the one-step 4DVAR. At the same time, both aspects of improvements are slightly increased with the increase of the number of processors, which shows better parallel scalability of the optimized version. This tendency disappears when the number of processors reaches 256 since the memory requirement for each process is not large enough compared with the local cache capacity. In addition, no extra memory is needed in

comparison with the underlying version. However, the peak memory for the underlying version on one processor reaches roughly 4.6 GB, whereas that for the optimized version decreases to roughly 4.0 GB.

The reliability of the optimized version has first been checked in terms of the adjoint correctness test in many applications, each of which is uniformly different by no more than two last significant digits in double precision. Next, many experiments from different applications are conducted using assimilation analysis to verify the smaller difference. Figure 6 shows the

declines of the cost function and its gradient when running the 4DVar system for both versions and the assimilation converges after 33 iterations. We can see that each curve of the optimized version perfectly agrees with the underlying one, which reveals that the analyses are identical for both versions.

Next, we increase the resolution of this case to the most often used domain of 270×180 horizontal grids and keep the 41 levels with the integration time step of 180 seconds. We calculate the ratios of running times between the adjoint model and

the nonlinear model with two different grids (Table 2). From these results, we can see that the improved performance of the adjoint model will be little affected by the model resolution when the number of processors is more than 4. Apparently, this ratio for both versions is only slightly increased with the increase of the number of processors for the two grids.

## 5 Conclusions

In this study, the conception of "Adjoint Locality" is intended to summarize a class of efficient methods and techniques

for obtaining required data for adjoint accumulations in any way. Specifically, the completely/partly removed sweep of the underlying function from adjoint subroutines/functions has the advantage of not only removing the structure dependency of the adjoint computational cost but also breaking down the fundamental assumptions of those traditional checkpointing methods.





Without much extra memory, an adjoint model of lower computational cost than its corresponding tangent linear model can be expected in this way. The modified push/pop-free method is also verified to be productive in reducing the cost of adjoint codes especially generated via TAPENADE or OpoenAD. Note that this method is remarkable if the adjoint cost is really lower enough, and much different within the same subroutine and across two different subroutines. In combining both

strategies, the running cost of the core adjoint procedure of WRF V3.7 in different applications is reduced by roughly half without an extra memory requirement. Across different numbers of processors, the time cost of WRF 4DVar from small- to large-scale grids declines by approximately 1/3 without losing any computational precision. Our experimental results also show that the improved performance of the adjoint model is never affected by the model resolution.

As the number of processors increases, the ratio of running time between the adjoint model and the nonlinear model also

increases, which shows that both the underlying version and the optimized version are of poor parallel performance. Most of our experiments show that the time cost of parallel communications during runs of the adjoint model is more than 1/3 when the number of processors is more than eight and that this percentage rapidly increases as the number of processors increases. Actually, the parallel communications before adjoint accumulations can be entirely removed from these adjoint subroutines if we push and pop required boundary data between the tangent linear model and the adjoint model. At the same time, much of

the adjoint calculations can be hidden in the parallel communications. In addition, we can expand the processor partition from 2 to 3 dimensions according to the calculation sequence of most WRF loops such that flexible partition can be selected for better parallel performance according to specific grid designs in different applications.

**Code availability**

Presented in the supplemen is the core adjoint subroutine solve_em_ad that has been modified basing on adjoint Locality and

the push/pop-free method. And the code located in the dyn_em file of WRFPLUSV3.7.

**Author contribution**

Cheng, Liu and Wang designed the experiments and Liu carried them out. Cheng developed the model code and Liu performed the simulations. Cheng and Liu prepared the manuscript with contributions from all co-authors.

**Acknowledgements**

We would like to thank WRFDA teams from NCAR for their instructive discussions and strong supports during this work.

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





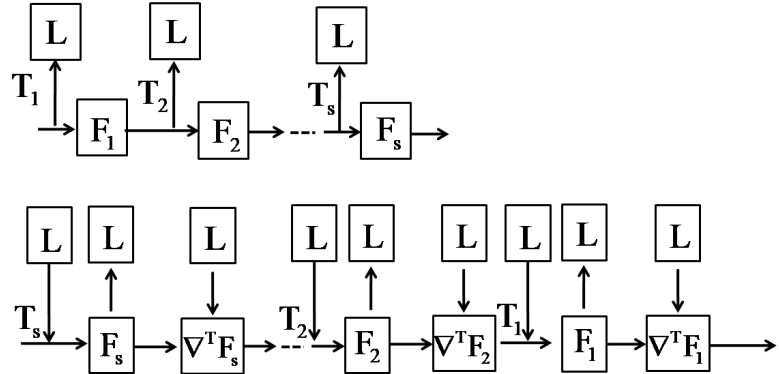

Figure 1: A typical push/pop strategy for adjoint implementation.



```
!Author of the underlying subroutine: M.E. Gorbu
!nov, revised fortran90 version.
Subroutine Adj_RefGr(Adj_X,X,Adj_Refr,Refr,
&Adj_Grad,Grad,NV)
    .....
!Calculations of the required variables through a
!full sweep of the underlying subroutine
  Call xyz2g (X, S, NV)
  Call LL2JK (S, P, NV)
  Call HLimits (P, Zint, Hmin, Hmax, NV)
  Call RefCIRA (X, RC, GC, NV)
  Call Int3SL(P, Zint, Ref, D2, Refr, RGrad, NV)
    .....
!Adjoint accumulations
  Do n=NV,1,-1
    Refr(n) =dexp(Refr(n))
      RGrad(n) =Refr(n)*RGrad(n)
      .....
```

```
      Adj_RGrad(n) =sin2*Adj_Grad(3,n)
      Adj_S(2,n)=dtr*cos2*RGrad(n)*Adj_Grad(3,n)
      .....
    End Do
!Recalculations of local required variables through
a
!small part of the underlying subroutine
  Call xyz2g(X,S,NV)
  Call LL2JK(S,P,NV)

  Call Adj_Int3SL(Adj_P,P,Zint,Ref,D2,Adj_Refr,
& Refr,Adj_RGrad,RGrad,NV)

  Call Adj_RefCIRA(Adj_X,X,Adj_RC,RC,
& Adj_GC,GC,NV)

  Call Adj_HLimits(Adj_P,P,Zint,Hmin,Adj_Hmax,
& Hmax,NV)

  Call Adj_LL2JK(Adj_S,S,Adj_P,P,NV)
  Call Adj_xyz2g(Adj_X,X,Adj_S,S,NV)
End Subroutine Adj_RefGr
```

Figure 2: An adjoint implementation with DFT/ADG.




```
!Revised version by hand. The corresponding push        Do n=NV,1,-1
!operations are located in the first run of RefGr.       !Remove expensive intrinsic function calculations
                                                         !with above flexible push/pop implementations
Subroutine Adj_RefGr(Adj_X,X,Adj_Refr,Refr,             !   Refr(n) =dexp(Refr(n))
& Adj_Grad,Grad,NV)                                          RGrad(n) =Refr(n)*RGrad(n)
 …..                                                         …..
!Remove 3 expensive subroutine callings with push/pop       Adj_RGrad(n) =sin2*Adj_Grad(3,n)
!operations across two different subroutines                Adj_S(2,n)=dtr*cos2*RGrad(n)*Adj_Grad(3,n)
!  Call xyz2g (X, S, NV)                                      …..
  CALL XPOPREAL8ARRAY(S,NV)                                 End Do
   Call LL2JK (S, P, NV)
   Call HLimits (P, Zint, Hmin, Hmax, NV)                 !Local recalculations can be removed that benefits
!  Call RefCIRA (X, RC, GC, NV)                           !from the above push/pop strategy
  CALL XPOPREAL8ARRAY(RC,MaxVec)                           !   Call xyz2g(X,S,NV)
  CALL XPOPREAL8ARRAY(GC,MaxVec)                           !   Call LL2JK(S,P,NV)
!  Call Int3SL(P, Zint, Ref, D2, Refr, RGrad, NV)          Call Adj_Int3SL(Adj_P,P,Zint,Ref,D2,Adj_Refr,
  CALL XPOPREAL8ARRAY(Refr,NV)                            & Refr,Adj_RGrad,RGrad,NV)
  CALL XPOPREAL8ARRAY(RGrad,MaxVec)                         …..
   …..                                                      Call Adj_xyz2g(Adj_X,X,Adj_S,S,NV)
!Adjoint accumulations, slightly revised                  End Subroutine Adj_RefGr
```

Figure 3: The revised adjoint implementation.



!*Both push and pop operations are located*
!*in the same adjoint subroutine. Here the*
!*push/pop functions are used as the same*
!*as the* TAPENADE *codes do.*

Subroutine Adj_A(….)
  …..
!*calculate the required values of* x
  CALL CALCULATE_DATA(….,
& x(1:N))
  …..
  CALL PUSHREAL8ARRAY(x,N)
  ……
  CALL POPREAL8ARRAY(x,N)
  …..
!*use the required value of* x *for local adjoint*
!*calculations*
  CALL DATA_USE(…, x(1:N), ….)
  …..
End Subroutine Adj_A

---

!*Push and pop operations are located in different*
!*subroutines. Here the subroutine* Ā *can be last run*
!*of the underlying subroutine* A, *or the tangent*
!*subroutine of* A, *or a sweep of* A *within adjoints.*

Subroutine Ā(….)
  …..
  CALL CALCULATE_DATA(…., x(1:N))
  …..
  CALL PUSHREAL8ARRAY(x,N)
  ……
End Subroutine Ā
Subroutine Adj_A
  …..
  CALL POPREAL8ARRAY(x,N)
  …..
  CALL DATA_USE(…, x(1:N), ….)
  …..
End Subroutine Adj_A

Figure 4: A push/pop implementation with AD softwares.



```
Subroutine Adj_A(….)
   …..
!keepx is a stack of length M*N allocated
!within or outside of subroutine A, where
!M is the number of times for pushing x.
!Here calculate the required values of x that
!is stored in the stack keepx directly.
   CALL CALCULATE_DATA(…., keepx(
& pointer:pointer+N-1))
   pointer =pointer+N
   ….
!  CALL PUSHREAL8ARRAY(x,N)
   ……
!  CALL POPREAL8ARRAY(x,N)
   …..
   pointer =pointer-N
   CALL DATA_USE(…, keepx(
& pointer:pointer+N-1), ….)
   ….
End Subroutine Adj_A
```

```
Subroutine Ā(….)
   …..
   CALL CALCULATE_DATA(….,x(1:N))
   …..
!  CALL PUSHREAL8ARRAY(x,N)
!push the required value of x into the stack keepx
   keepx(pointer:pointer+N-1) =x(1:N)
   pointer =pointer+N
   ……
End Subroutine Ā

Subroutine Adj_A(….)
   …..
!  CALL POPREAL8ARRAY(x,N)
   pointer =pointer-N
   CALL DATA_USE(…, keepx(
& pointer:pointer+N-1), ….)
   …..
End Subroutine Adj_A
```

Figure 5:The revised adjoint implementation through the push/pop free method.



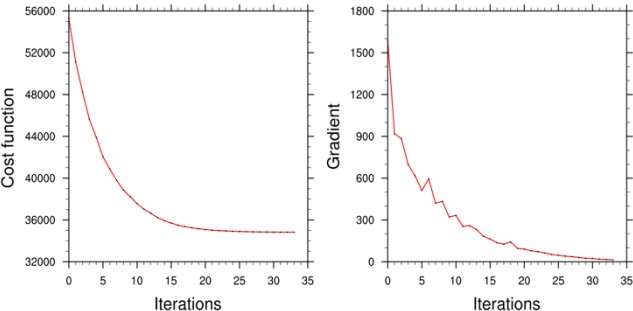

Figure 6: The cost function and its gradient for running the underlying version (red line) and the optimized version (black dots).





Table 1. Running time for one-time-step integration with WRFDA 3.7 on 1-256 processors. Units: second; grids: 96×60×41.

| NP | | 1 | 2 | 4 | 8 | 16 | 32 | 64 | 128 | 256 |
|---|---|---|---|---|---|---|---|---|---|---|
| **Nonlinear Model** | | 0.96 | 0.502 | 0.277 | 0.167 | 0.112 | 0.061 | 0.038 | 0.026 | 0.025 |
| **TL Model** | | 1.18 | 0.61 | 0.350 | 0.236 | 0.188 | 0.102 | 0.068 | 0.048 | 0.042 |
| **Adjoint Model** | Underlying | 3.20 | 1.66 | 0.935 | 0.680 | 0.527 | 0.328 | 0.217 | 0.160 | 0.117 |
| | Optimized | 1.78 | 0.92 | 0.518 | 0.367 | 0.291 | 0.166 | 0.108 | 0.077 | 0.063 |
| | Improved | 44.4% | 44.6% | 44.6% | 46.0% | 44.8% | 49.4% | 50.2% | 51.9% | 46.2% |
| **4DVar System** | Underlying | 9292 | 4822 | 2734 | 1952 | 1523 | 920 | 615 | 449 | 355 |
| | Optimized | 6342 | 3279 | 1864 | 1298 | 1032 | 584 | 390 | 288 | 244 |
| | Improved | 31.7% | 32.0% | 31.8% | 33.5% | 32.2% | 36.5% | 36.6% | 35.9% | 31.3% |



Table 2. Time cost ratios between the adjoint model and the nonlinear model for the underlying version and the optimized version.

| Grids | NP | 1 | 2 | 4 | 8 | 16 | 32 | 64 | 128 | 256 |
|---|---|---|---|---|---|---|---|---|---|---|
| 96×60×41 | Underlying | 3.33 | 3.31 | 3.38 | 4.07 | 4.71 | 5.38 | 5.71 | 6.15 | 4.68 |
| | Optimized | 1.85 | 1.83 | 1.87 | 2.20 | 2.60 | 2.72 | 2.84 | 2.96 | 2.52 |
| 270×180×41 | Underlying | 4.12 | 4.05 | 4.10 | 4.41 | 4.56 | 4.74 | 4.96 | 5.68 | 5.13 |
| | Optimized | 2.30 | 2.29 | 2.31 | 2.52 | 2.73 | 2.74 | 2.82 | 2.88 | 2.75 |