# Peer review of "Optimization of the WRFV3.7 adjoint model"

_Geoscientific Model Development, 2018_

## Referee Comment (RC1) · Anonymous Referee #1 · 1 Jan 2019

This paper proposes code optimizations and strategies to improve the performance of adjoint models in WRF. The code has been already merged as part of WRFPLUS.

**1 General comments:**

- Overall, from a computational perspective, I see little evidence or data supporting your conclusions. I would suggest to add more information about your analysis and verification methods. (Please see my Specific Comments).

- You mention that your code is already part of WRFPLUS (dyn_em). Can you please add any link or URL for a pull request or discussion of the code? This is valuable information for future work and other scientists.

- English: Can you please make a general review, readability could be improved and some phrases and expressions are hard to understand.

**2 Specific Comments:**

- P2, l16: "Apparently, different ways have different costs in runtime or memory." Can you please be more specific (examples, including data about impact on memory usage).

- P2, l21: You make a leap from WRF to WRFPLUS that can make it confusing for readers that are not familiar with the version (package) differences. Can you please add some text in here (1/2 lines) talking about their differences thus, increasing the readability?

- P3, l5: "This implementation requires comparatively less memory consumption but much more computational cost, although there are always productive ways available for reducing the latter in practice." Can you please be more specific on about 'less memory' and 'more computational cost'? Can you please give an example of other ways of resources usage optimization?

- P3, l31: "As a typical strategy of the reverse accumulations..." Can you please add a reference in here?

- P6, l25: "allocating/ deallocating them outside of the running cycles of the procedures." Can you please specify the cost of this? What it is the benefit of this for the overall run (intuitively this will improve performance on your part of the code but move the cost somewhere else).

- P7, l23: "Through careful IO analysis". During all the paper you did not indicate any technique (e.g. tracing, or data dumping) that lead you to all these

conclusions more than reading the code and finding IO calls. Can you please elaborate this more and provide more insights/evidence of this analysis and its conclusions?

- P8, l7: "on a cluster system with 250 nodes, each of which has 20 processors" Can you please be more specific (node type, processor type, memory, network interconnection...)?

---

## Referee Comment (RC2) · Anonymous Referee #2 · 7 Mar 2019

This manuscript present a refactoring of some routines of WRFPLUS in order to decrease the execution time maintaining the total memory required.

**1  General comments**

- Authors state that they have used several techniques (push/pop-free method, IO analysis, use of adjoint locality) but my first impression is that the paper consist only on a code refactoring affecting some routines. This refactoring was basically done replacing superfluous subroutine calls with push/pop operations and modifying the way that push/pop operations are done. Even if they obtain a significant reduction in the execution time, I do not see in this work a real scientific contribution, but rather an engineering practice.

- The paper is difficult to read and understand. A general review in order to improve the quality of the writing is necessary.

**2  Specific comments**

- About the Input/output analysis. Usually, I/O refers to disk or network accesses, but no memory accesses. In the paper it is difficult to see if a real I/O analysis has been carried out.

- P2, L6-7: "To some AD applications such as 4DVAR, much more memory consumption must result in poor scalability both in computing grids and in different computer environments." Why is this true? What do you mean by "computing grids"?

- P2, L31-32: "...by a slight of hand several years ago", What does this expression mean?

- P2-3, L34-1: "Actually, the computational cost for these push/pop operations will turn out to be comparatively expensive when the local adjoint cost is reduced at a lower degree." I suppose that you are talking about the percentage of the push/pop operations over the total cost, aren't you? Please clarify it.

- P3, L8-10: "...the most difficult optimization may be associated with huge memory consumption stemming from the fact that the computational cost of the core adjoint procedure is almost evenly distributed across its calling subroutines..." Why is the memory consumption related with the cost balance of the subroutines? Please, clarify.

- P3, L16: "global/local IO", What do you mean by global and local IO in this context?

- P3, L26. You should specify that $X$ and $Y^*$ are in this context.

- P4, L18 and Fig. 1: Figure 1 does not help to understand the concept of "adjoint locality". A better explanation of this figure is necessary.

- P5, 2nd paragraph: It is said that required data are pushed into stack "...during each sweep of RP when running its adjoint subroutine Adj_RP...". In the same paragraph: "However, the sweep of RP within Adj_RP will be completely removed for better adjoint performance..." If the sweep of RP is removed, how are data pushed into the stack? Please, clarify.

- Figure 3: How can the second call to LL2JK be removed if P is not popped out from the stack?

- P5, L16: Please, indicate what NV is.

- P6, L19-21: "First, we use several or more stacks instead of only one, each of which is still a 1D data structure. It has been shown that this type of data structure has the advantages of smaller access cost and flexible expressions, either in the communications between procedures or in local program calculations." Can you please add any reference to justify this assertion?

- P6, L23. "Although the cost of allocating/deallocating many dynamical stacks cannot be neglected compared with the costs of these adjoint procedures..." Again, this assertion should be justified. Which memory allocator are you using?

- Section 3. In the paper, push/pop operations are replaced by copying the data into a vector (keepx) and then recovering them from that vector. I cannot see how different this data movement is from using a stack. In the paper it is said (P7, L14-16) "Note that the pushing operation PUSHREAL8ARRAY is replaced by a direct evaluation statement from x to keepx with approximately the same cost both in

run time and in memory." So the only improvement is due to the replacement of the pop operation. Might not be simpler to optimize the use of this operation?

- P7, L15: "...a direct evaluation statement from x to keepx..." Please, clarify what do you mean by "a direct evaluation", isn't it a simple copy?

- P7, L28. "sulfurous calculations." I suppose you mean "superfluous".

- Regarding the test results, more details about how the experiments were carried out should be included, e.g., compiler used, optimization level, MPI version, characteristics of the cluster (cores per processor, memory per core, etc).

- Table 1: You use until 256 processors, in a system with 5000 processors (250*20). It will be interesting to see values for a larger numbers of PEs and a larger resolution.

- Table 1: How many experiment have been done for each value of PE? Are the results in the table an average? What is the standard deviation?

- P8, L11-12: "At the same time, both aspects of improvements are slightly increased with the increase of the number of processors..." What are those "both aspects"?

- P8, L18: "...uniformly different by no more than two last significant digits in double precision." If you do not mean the last two bits in the double precision binary representation, this statement is meaningless. You should use the absolute difference between values.

- Test result section: It will be very interesting add some graph to show whether any change in scalability (both strong and weak) is caused by the proposed improvements.

---

## Author Comment (AC1) · 23 Apr 2019

Dear reviewers

First of all, thank you very much for reviewing our manuscript. Per your comments (in black font), we have revised our manuscript accordingly (in red font) and made point-to-point responses (in blue font) to all the comments and concerns. Below are our detailed responses.

5

Reviewer#1

**1 General comments:**

- Overall, from a computational perspective, I see little evidence or data supporting your conclusions. I would suggest to add more information about your analysis and verification
10 methods. (Please see my Specific Comments).

Reply: We very appreciate the helpful suggestions, although most of the manuscript are discussions of code optimization techniques.

- You mention that your code is already part of WRFPLUS (dyn_em). Can you please add any link or URL for a pull request or discussion of the code? This is valuable information
15 for future work and other scientists.

Reply: Sorry to fail to download the complete codes. We move the complete code to a new site (https://github.com/juanjliu/WRFDA-OPTIMAZATION  with a simple readme as following:

Step 1: Download the underlying versionV3.7

20 Users can download the WRFPLUS source code from http://www2.mmm.ucar.edu/wrf/users/wrfda/download/wrfplus.html , then Unzip and untar the WRFPLUS file.

Step 2: Download the optimization code

User can download the optimization code from this web:
25 https://github.com/juanjliu/WRFDA-OPTIMAZATION.git

Step 3: Unzip and untar the WRFDA-OPTIMAZATION file

cd WRFPLUSV3    (from Step 1)

Using optimization codes to Update codes in the underlying version

Step 4: Installing WRFPLUS for 4D-Var Run

30 cd WRFPLUSV3

./configure wrfplus

./compile em_real >& compile.out

More detail information can be found from "README.md" on
https://github.com/juanjliu/WRFDA-OPTIMAZATION.git.

35 • English: Can you please make a general review, readability could be improved and some phrases and expressions are hard to understand.

Reply: The constructive comments are highly appreciated. In fact, we have done two polishing services. To improve the readability, we further modify some phrases and expressions in the manuscript (marked in red font).

40 **2 Specific Comments:**

• P2, l16: "Apparently, different ways have different costs in runtime or memory." Can you please be more specific (examples, including data about impact on memory usage).

Reply: Thanks for your suggestion. However, this is a big problem to answer since there are innumerable ways for adjoint implementations. For this reason, we can present several documents and discussions in this manuscript.

And here, we present some ways as examples:

Since the adjoint accumulation is always performed in a reverse way ( $\nabla^T F_s \rightarrow \nabla^T F_1$ See Fig 1 in the manuscript), we face the problem: how can the value of a required variable (generally called basic states $T_1, T_2, \cdots, T_s$) for the current adjoint calculation be obtained. Generally, there are some ways to do including directly restoring and directly recalculating.

For directly restoring, one can use Output/Input (OI) Memory or OI Disk (See http://www2.mmm.ucar.edu/wrf/users/wrfda/Docs/user_guide_V3.7/users_guide_chap6.htm#_Description_of_Namelist_2).

OI Memory: one can successively push the required data $T_1, T_2, \cdots, T_s$ into a huge stack L in the last run of the underlying functions and naturally pop them out in the reverse adjoint accumulations one by one. Then OI Memory will need huge costs in memory.

OI Disk: The required data $T_1, T_2, \cdots, T_s$ was written into disk one by one. It has little memory-consuming, but has expensive time-consuming so the WRFDA would not support the OI Disk option.

For directly recalculating, one can recalculating it from the start of the underlying functions, or calculating it from intermediate results. The former has more expensive time-consuming than the latter.

65 • P2, l21: You make a leap from WRF to WRFPLUS that can make it confusing for readers that are not familiar with the version (package) differences. Can you please add some text in here (1/2 lines) talking about their differences thus, increasing the readability?

Reply: Sorry for confusion here. The Weather Research and Forecasting (WRF) model is a numerical weather prediction and atmospheric simulation system designed for both research and operational applications. The adjoint model and the tangent linear model (called WRFPLUS) based on WRF are developed for the four-dimensional variational data assimilation (4DVar), cloud analysis, forecast sensitivity to observations, and chemistry data assimilation et al. (Zhang et al. 2013).

75 For clarification, we revised manuscript.

Reference

Zhang, X., Huang, X. Y. and Pan, N.: Development of the Upgraded Tangent Linear and Adjoint of the Weather Research and Forecasting (WRF) Model. J. Atmos. Ocean. Tech., 30(6), 1180-1188, doi: 10.1175/jtech-d-12-00213.1, 2013.

- P3, l5: "This implementation requires comparatively less memory consumption but much more computational cost, although there are always productive ways available for reducing the latter in practice." Can you please be more specific on about 'less memory' and 'more computational cost'? Can you please give an example of other ways of resources usage optimization?

  Reply: "Different with the saving-all strategy as TAPENADE, this implementation of a 2-level checkpointing structure requires comparatively less memory consumption but much more computational cost, although there are always productive ways available for reducing the latter as shown in next discussions."

- P3, l31: "As a typical strategy of the reverse accumulations..." Can you please add a reference in here?

  Reply: Done. Thanks.

- P6, l25: "allocating/ deallocating them outside of the running cycles of the procedures." Can you please specify the cost of this? What it is the benefit of this for the overall run (intuitively this will improve performance on your part of the code but move the cost somewhere else).

  Reply: In Fortran 90/95 programs, the dynamic array allocation within tight loops can really slow down the execution of my code.

  i.e., several medium-sized arrays are allocated inside a loop, like:

  Do i=1,1000

  Allocate(tmp(20))

  tmp(1:20)=1d0

  call foo(tmp)

  deallocate(tmp)

  enddo

  But in WRFPLUS, "it"is really of a very small cost for the overall run "in some cases, we can significantly reduce "it" by allocating/deallocating them outside of the running cycles of them", we will change the statements to make it clear. Thanks.

- P7, l23: "Through careful IO analysis". During all the paper you did not indicate any technique (e.g. tracing, or data dumping) that lead you to all these conclusions more than reading the code and finding IO calls. Can you please elaborate this more and provide more insights/evidence of this analysis and its conclusions?

Reply:   IO analysis is a relative notation that is employed to describe the input/output behavior of a variable within a program object defined by a segment of program lines. By recording the IO knowledge of argument parameters in each procedure, it can calculate the final IO relationship of any argument parameter through deep recursive dependence analysis. This process can be represented as an iteration $A_{k+1} = A_k \oplus A_k^2$, where $A_0$ is the initial dependence matrix.

Because IO technique has been discussed in other documents, so we add two references as "Through careful IO analysis and the "To be recorded" analysis techniques for reverse mode (Cheng et al. 2009a, Hascoet et al. 2005)" in the revised manuscript. Thanks.

Reference:

Cheng Q., H.B. Zhang and Y.H. Zhao: Differentiation Transforming System, Nature Sciences Progress, 19(3): 397-406, doi: 10.1016/j.pnsc.2008.07.012 , 2009a.

Hascoet, L., U. Naumann and Pascual, V.: "To be recorded" analysis in reverse-mode automatic differentiation. Future Gener. Comp. Sy., 21(8). 1401-1417, doi: 10.1016/j.future.2004.11.009 ,2005.

- P8, l7: "on a cluster system with 250 nodes, each of which has 20 processors" Can you please be more specific (node type, processor type, memory, network interconnection...)?

Reply: Thanks for your suggestion. We have added it in the revised manuscript.

---

## Author Comment (AC2) · 23 Apr 2019

Dear editors and reviewers

Thanks a lot for your scores of suggestions about this manuscript. Per your comments (in black font), we have revised our manuscript accordingly (in red font) and made point-to-point responses (in blue font) to all the comments and concerns. Below are our detailed responses.

Referee #2,

This manuscript presents a refactoring of some routines of WRFPLUS in order to decrease the execution time maintaining the total memory required.

Reply: Sorry, no complete code is uploaded during the first manuscript. Here, we move the complete code to a new site (https://github.com/juanjliu/WRFDA-OPTIMAZATION).

More detail information about the usage can be found from "README.md" on https://github.com/juanjliu/WRFDA-OPTIMAZATION.git.

The complete code is helpful to understand "Refactoring is based on several techniques and full using of the conception "adjoint locality"".

1 General comments

Authors state that they have used several techniques (push/pop-free method, IO analysis, use of adjoint locality) but my first impression is that the paper consists only on a code refactoring affecting some routines. This refactoring was basically done replacing superfluous subroutine calls with push/pop operations and modifying the way that push/pop operations are done. Even if they obtain a significant reduction in the execution time, I do not see in this work a real scientific contribution, but rather an engineering practice.

Reply: We just say we could do nothing about these code optimizations without the direction of the conception "adjoint locality". And how valuable is the push/pop-free method directly decided from practice instead of anything else.

In the example shown in Figure 2, there is not any superfluous subroutine calls without our consideration with adjoint locality. Generated the adjoint code of a subroutine/function with most AD tools including TAPENADE, a full sweep/duplication is always required before local adjoint accumulating, which is clearly shown in Figure 1. And this is the fundament fact of some checkpointing methods. However, such weep is not always indispensable to adjoints in a sense of adjoint locality.

We present all modified code, you will find that we just purely using the push/ pop-free method within the core adjoint subroutine solve_em_ad. Within the adjoint code optimization of WRF, we scarcely used such replacement which is extracted from another application the GPS/MET rayshooting model as shown in Figure 2. However, we in most cases decrease most

execution time in most adjoint routines just by changing its computational sequence with the help of the understanding of adjoint locality.

Actually, the word "SCIENCE" is so noble instead of anything with only a few modifications or several skills. We never thought there was scientific contribution in this article, but we do believe the techniques and the conception discussed are really valuable in automatic differentiation fields. And we were told that new techniques and new methods for code improvement were welcomed by GMD. If you compare our results with those from other AD fellows, you will find that it is so difficulty to require such less adjoint execution cost without any/extra memory increment at the same time.

The paper is difficult to read and understand. A general review in order to improve the quality of the writing is necessary.

Reply: English is really a big problem to us although we have tried to make it clear so many times for readers. We wish there is a fellow who is interested in this article from English-speaking country to join in the author list.

**2 Specific comments**

• About the Input/output analysis. Usually, I/O refers to disk or network accesses, but no memory accesses. In the paper it is difficult to see if a real I/O analysis has been carried out.

Reply: I agree with you. IO here refers to "Input and Output" of a program object such as a subroutine, which has clearly defined in the abstract part.

• P2, L6-7: "To some AD applications such as 4DVAR, much more memory consumption must result in poor scalability both in computing grids and in different computer environments." Why is this true? What do you mean by "computing grids"?

Reply: "computing grids" means the resolution of model. For example, the model state $x_i$ is a solution of model equations:

$$\forall i, \qquad x_i = M_{0 \to i}(x_0)$$

where $M_{0 \to i}$ is a predefined model forecast operator from the initial time to $i$. For the second case in the manuscript, we use 270×180 horizontal grids and keep the 41 levels, focus on 5 model variables and the integration time step of 180 seconds, then the dimension of the basic state ($x_i$) at one step is 270×180×41×5. If the forecast time is 12 hours, there will be 240 steps ($x_i$) to save. For example, WRFDA successively pushes the required data $x_1, x_2, \cdots, x_i$ into a huge stack L, then it needs huge costs in memory.

So, as the number of computing grids increases, the extra memory for running the adjoint model could be out of the available memory.

• P2, L31-32: ". . . by a slight of hand several years ago", What does this expression

mean?

Reply: It should be "with a slight of revision". Thanks.

• P2-3, L34-1: "Actually, the computational cost for these push/pop operations will turn out to be comparatively expensive when the local adjoint cost is reduced at a lower degree." I suppose that you are talking about the percentage of the push/pop operations over the total cost, aren't you? Please clarify it.

Reply: The computational cost of these push/pop operations are negligible when there is no accompanying adjoint optimization, but if the adjoint model is optimized and the ratio between the adjoin model and Nonlinear model reduces from 5:1 or 6:1 to 3:1 or even lower, the percentage of the push/pop operations can't be ignored.

For example, you cut 1 from 10 and you can get 10% improvement. However, if you cut 1 from 3 or less and you can get more than 33% improvement)

• P3, L8-10: ". . . the most difficult optimization may be associated with huge memory consumption stemming from the fact that the computational cost of the core adjoint procedure is almost evenly distributed across its calling subroutines. . . " Why is the memory consumption related with the cost balance of the subroutines? Please, clarify.

Reply: "which resulting in scores to a hundred of push/pop operations inserted between these calling subroutines and their adjoints are required within each call of the core adjoint procedure for less execution time." is added in the revised manuscript. However, an opposite case such as presented in Figure 2 requires less memory, in which the computational cost of the considered subroutine is concentrated on one or only a few calling subroutines.

• P3, L16: "global/local IO", What do you mean by global and local IO in this context?

Reply: A variable may have different IO attribute during different program objects as well as different local segment of program lines.

• P3, L26. You should specify that X and Y _ are in this context.

Reply: Thanks a lot. "where $\delta X$ and $\delta Y^*$ are respectively the input of both differential models." is added in the revised manuscript.

• P4, L18 and Fig. 1: Figure 1 does not help to understand the concept of "adjoint locality". A better explanation of this figure is necessary.

Reply: Yes. Figure 1 is only a typical push/pop strategy for most adjoint implementations from practical applications through which we present our basic ideas about "adjoint locality", we add a general example of subroutine/function splitting for this suggestion.

• P5, 2nd paragraph: It is said that required data are pushed into stack ". . . during each sweep of RP when running its adjoint subroutine Adj_RP. . . ". In the same paragraph: "However, the sweep of RP within Adj_RP will be completely removed for better adjoint performance. . . " If the sweep of RP is removed, how are data pushed into the stack? Please, clarify.

Reply: It is clear to show this from the last sentence "In Figure 3, we therefore only keep a very small part of the required data directly from the first run of the root subroutine RayFind with the extra memory…."

• Figure 3: How can the second call to LL2JK be removed if P is not popped out from the stack?

Reply: That is because Int3SL calling has been removed in Figure 3 such that the value of P was never changed not as in Figure 2.

• P5, L16: Please, indicate what NV is.

Reply: Yes, Thanks.

• P6, L19-21: "First, we use several or more stacks instead of only one, each of which is still a 1D data structure. It has been shown that this type of data structure has the advantages of smaller access cost and flexible expressions, either in the communications between procedures or in local program calculations." Can you please add any reference to justify this assertion?

Reply: Thank you very much for your suggestions. But in fact, we just speak this from our practice with WRFPLUS. Based on careful consideration, we add "in practice" into this statement Instead of adding a reference.

• P6, L23. "Although the cost of allocating/deallocating many dynamical stacks cannot be neglected compared with the costs of these adjoint procedures. . . " Again, this assertion should be justified. Which memory allocator are you using?

Reply: "…allocating/deallocating many dynamical stacks in Fortran 90/95 programs…" is added in the revised manuscript. Thanks.

• Section 3. In the paper, push/pop operations are replaced by copying the data into a vector (keepx) and then recovering them from that vector. I cannot see how different this data movement is from using a stack. In the paper it is said (P7, L14-16) "Note that the pushing operation PUSHREAL8ARRAY is replaced by a direct evaluation statement from x to keepx with approximately the same cost both in run time and in memory." So the only improvement is due to the replacement of the pop operation. Might not be simpler to optimize the use of this operation?

Reply: Several similar questions have been asked. Thanks. Therefore we add something such as "Actually, the computational cost for these push/pop operations will turn to be comparatively expensive in some cases such as when the push and pop operations are located in different innermost loops or if the local adjoint cost is reduced at a lower degree." to the introduce part. 1) Above says the cost of push/pop operations cannot be overlooked in some cases. 2) Only the pop operation is saved in corresponding to this case when push and pop operations are located in different subroutines or functions, the past answer is "Statistically both the push and the pop operations can be improved in this way in most adjoint codes. Less adjoint implantations use a pair of push/pop operations in different subroutines/functions." And we have presented "Another case is **infrequently** used in some adjoint implementations that the push operation and the corresponding pop operation are located in two different subroutines or functions in" in the last part of Section 3. 3). It is really not easy to use this method everywhere

such that we actually have revised only less than 5% of the total push/pop operations in those adjoint subroutines/functions. However, that is enough to have an obvious improvement of the adjoint model.

• P7, L15: ". . . a direct evaluation statement from x to keepx. . . " Please, clarify

what do you mean by "a direct evaluation", isn't it a simple copy?

Reply: Yes, it is. That means only the pop operations can be saved in such cases. "…a simple duplication from x to keep_x…."

• P7, L28. "sulfurous calculations." I suppose you mean "superfluous".

Reply: Yes, Thanks.

• Regarding the test results, more details about how the experiments were carried out should be included, e.g., compiler used, optimization level, MPI version, characteristics of the cluster (cores per processor, memory per core, etc).

Reply: All experiments were conducted on LINUX and a Distributed Memory (DM) server cluster with 250 nodes connected with InfiniBand, each of which has 20 Intel Xeon E5-2670/2.5G processors and shares 62 GB memory. Compiler used is Intel-ifort version 13.1.1, optimization level is O3, and mpiifort for the Intel(R) MPI Library 4.1 for Linux. All information have been added in the revised manuscript.

We will upload the configure as a supplementary material.

• Table 1: You use until 256 processors, in a system with 5000 processors (250*20). It will be interesting to see values for a larger numbers of PEs and a larger resolution.

Reply: WRFPLUS will lose a lot of parallel performance for practical applications when the number of processors is more than 512. And we scarcely have chance for using more than 512 processors at present.

• Table 1: How many experiment have been done for each value of PE? Are the results in the table an average? What is the standard deviation?

Reply: This is a big problem. Each data of the time cost for the core adjoint procedure solve_em_ad in Table 1 and Table 2 is picked up from scores to hundreds of data, almost an average of those data. For example, the core adjoint procedure solve_em_ad needs to execute for scores of times within each calculation of gradient, but we just select one that is probably most near to the average. So we revise it into "Table 1 shows the **average** wall-clock time for one-time-step integration….."

• P8, L11-12: "At the same time, both aspects of improvements are slightly increased with the increase of the number of processors. . . " What are those "both aspects"?

Reply: Thanks. So "…both aspects of improvements for the adjoint model and the 4DVar system…"

• P8, L18: ". . . uniformly different by no more than two last significant digits in double

precision." If you do not mean the last two bits in the double precision binary representation, this statement is meaningless. You should use the absolute difference between values.

   Reply: The correctness of the adjoint model can be checked by the following algebraic expression:

$$(M_r z)^T (M_r z) = z^T (M_r{}^T (M_r z)) \quad (R1)$$

Where $M_r$ is the tangent linear model of nonlinear model, $M_r{}^T$ is the adjoint model. If the code is correct, equation R1 should be verified for all input values of $z$ . Compare the values of left and right to see whether they are equal to the machine accuracy. 13 digit accuracy is the best result to be expected. In some rare cases, less than 13 digit accuracy can still represent a none-error adjoint code. A lot of experiments show that a minimum of 9 digit accuracy are required for a single-precision machine (see NCAR Technical Note, 1997, NCAR/TN-435-STR). For example, our checking result is the following:

VAL_L:      0.23958449830151E+10

VAL_R:      0.23958449830152E+10

I agree with that it is not clear here. So, we revised it into "The reliability of the optimized version has first been checked in terms of the correctness test **(Zhang et al. 2013)** in many applications, each of which is uniformly different by no more than two last significant digits in double precision **with the underlying version**."

Reference

Zhang, X., Huang, X. Y. and Pan, N.: Development of the Upgraded Tangent Linear and Adjoint of the Weather Research and Forecasting (WRF) Model. J. Atmos. Ocean. Tech., 30(6), 1180-1188, doi: 10.1175/jtech-d-12-00213.1, 2013

• Test result section: It will be very interesting add some graph to show whether any change in scalability (both strong and weak) is caused by the proposed improvements.

   Reply: Thanks. But in some cases, it is very difficult to clarify a change is whether using the push/pop-free method or other techniques in terms of adjoint locality. So for this comment, we believe that there will be a lot of extra work to do.